# Mediating roles of perceived stigma and mental health literacy in the relationship between school climate and help-seeking behavior in Indonesian adolescents

**Mohammad Hendra Setia Lesmana[1,2], Min-Huey Chung[1,2] \***

**1** School of Nursing, College of Nursing, Taipei Medical University, Taipei City, Taiwan, **2** Department of Nursing, Shuang Ho Hospital, Taipei Medical University, New Taipei City, Taiwan

\* minhuey300@tmu.edu.tw

## Abstract

### Objective

This study aimed to investigate whether perceived stigma and mental health literacy play mediating roles in the correlation between school climate and help-seeking behavior in Indonesian adolescents.

### Methods

We used cross-sectional study design that recruited 760 Indonesian adolescents of age 16–19 years. We used convenience sampling from July to September 2019. Bivariate analysis was used to investigate the association of demographic characteristics with help-seeking behavior. Mediation analysis was employed to explore the mediating roles of mental health literacy and perceived stigma on the relationship between school climate and help-seeking behavior.

### Results

Findings indicated that ethnicity, family income, and father's educational level were significantly associated ($p < 0.05$) with help-seeking behavior in Indonesian adolescents. Furthermore, perceived stigma and mental health literacy sequentially showed partial mediating roles in the relationship between school climate and help-seeking behavior (indirect effect: 0.004; 95% CI: 0.001, 0.010). Our mediating model indicated that a high level of school climate was associated with low perceived stigma ($b = -0.11$, $p < 0.001$) and high mental health literacy ($b = 0.28$, $p < 0.001$) and higher help-seeking behavior ($b = 0.14$, $p < 0.001$).

### Conclusion

Our study discovered novel insight of help-seeking behavior mechanism among adolescent by serial mediation test. Supportive school climate is necessary to achieve adequate help-

**Data Availability Statement:** All relevant data are within the paper and its Supporting information files.

**Funding:** This work was supported by grants from the University System of Taipei Joint Research Program 【USTP-NTPU-TMU-113-02】.

**Competing interests:** The authors have declared that no competing interests exist.

seeking behavior. In addition, taking into account of student's perceived stigma and mental health literacy in promoting help-seeking behavior is also important.

## Introduction

Mental health problems have become a global burden, particularly among adolescents [1]. The occurrence of mental health problems was highest among individuals in the 16–24 age group compared to any other period in life [2]. Mental health problems were also frequently observed during childhood and adolescence, impacting about 14% of individuals aged 4 to 17 years [3]. Approximately, 1 in 10 adolescents was affected by mental health problems, including anxiety, depression, and personality disorder [4]. In Indonesia, the Ministry of Education, Culture, Research, and Technology (KEMENDIKBUD) implemented Policy Number 111 in 2014 in response to the growing concerns about mental health issues among adolescents [5]. This policy mandates that every school should establish a dedicated department or section focused on guidance and counseling to promote the overall well-being of their students [5]. However, according to the ASEAN mental health system report in 2016, a significant number of mental health problems were prevalent among adolescents in Indonesia [6]. The 2013 Indonesia Basic Health Research (RISKESDAS) reported that mental health problems were accounted by 11.3% in East Java, Indonesia [7]. In addition, around 70% of adolescents suffering from mental health problems did not receive appropriate treatment immediately [4]. The World Health Organization (WHO) reported that 10% to 20% of adolescents encountered by mental health problems remained underdiagnosed and undertreated [8]. Meanwhile, a study mentioned that 35% of adolescents who experiencing emotional and mental health problems did not tend to seek help [9]. Furthermore, only one in three adolescents with mental health problems sought for help [10]. Mental health problems among adolescents contributed to suicide and self-harm that ranking as the fourth leading cause of mortality in the world [8], as well as, the top 10 cause of disabilities in East Java, Indonesia [11]. Therefore, we conclude that mental health problems in adolescents is a serious concern, particularly, regarding help-seeking behavior among adolescents.

According to reasoned action theory, people's behavior is expected to be rational, consistent, and frequently automatic, based on their beliefs regarding the behavior they are performing [12]. There are three factors affected people's behavior [12]. First, background factors related information such as knowledge. Second, people's beliefs including behavioral belief such as stigma that followed by attitude toward the behavior. Third, the behavior can be directly influenced by factors of actual control, such as the environment. The existence of environmental limitations can hinder individuals from engaging in specific behavior including help-seeking behavior [12]. Based on studies, adolescents in Asia spend most of their time at school, which makes school has pivotal roles in maintaining adolescents mental well-being [13–15]. In this regard, the school climate was pronounced highly associated with help-seeking behavior. The supportive school environment from teachers, peers, and safety environment increased the willingness of adolescents to express their concern and accessing help [16–18]. In addition, positive school climate (i.e., safety, engagement and environment aspects) engaged adolescents to increase literacy and decrease stigma about mental health problem [19]. Moreover, prior studies suggested that perceived or internal stigma on mental health problem would inhibit adolescents to seek for help [9, 20, 21]. On the other hand, higher mental health literacy level encouraged adolescents seeking for help [22]. Based on aforementioned

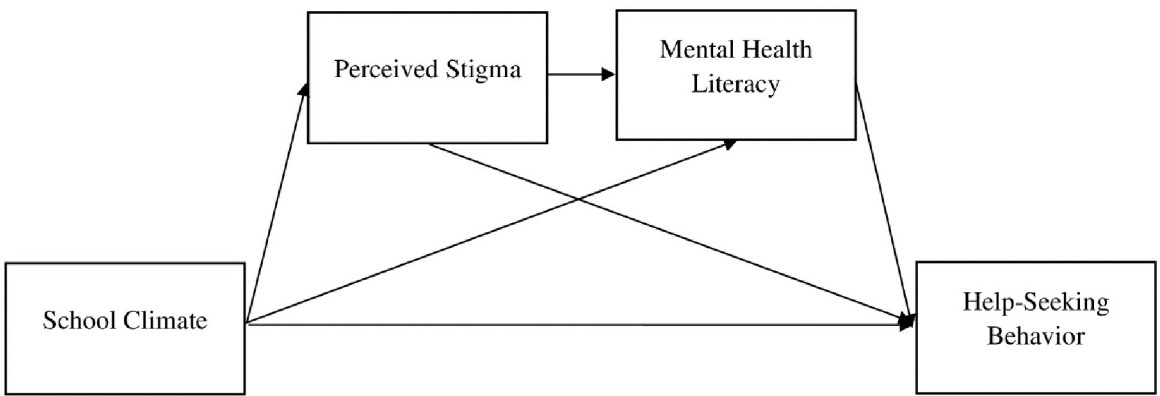

**Fig 1. Hypothesis framework based on reasoned action model and related literatures.** There was sequential mediating effect of perceived stigma and mental health literacy on relationship between school climate and help-seeking behavior.

explanation, the association among environment (school climate), knowledge (mental health literacy), behavioral belief (perceived stigma), and behavior (help-seeking behavior) need to be explored further (Fig 1).

To the best our knowledge, there was no study conducted to explore the link among school climate, mental health literacy, perceived stigma, and help-seeking behavior in adolescents. Furthermore, lacking study related help-seeking behavior was conducted in Indonesian adolescents. Thus, our study aimed to analyze the mediating effect of mental health literacy and perceived stigma on the relationship between school climates and help-seeking behavior in Indonesian adolescents.

## Methods

### Participants and procedure

This study enrolled the participants from four public senior high schools (i.e., SMAN 1 Gading, SMAN 1 Kraksaan, SMAN 1 Probolinggo, SMAN 1 Gending), which located in East Java, Indonesia. The convenience sampling method was performed to collect the participants from July to August 2019. The inclusion criteria were students who completed the written informed consent form and obtained permission from their parents or legal representative; who could write and read Indonesian; who were aged 16–19 years; and who were in grade 10–12[th].

Students who were undergoing treatment for mental health disorders or were absent from school during data collection were excluded from the study. The sample size calculation was conducted using G*Power software version 3.1.9.4 [23]. The sample size was calculated under statistical test of multiple linear regression analysis with 12 predictors, including demographic variables, school climate, perceived stigma, and mental health literacy. The effect size of 0.15 [24], Cronbach's alpha of 0.05, and a power of 0.95 were inputted. the sample size was calculated equals to 184 participants. After considering the dropout rate of 20%, this study at least recruited 184+37 = 221 participants. However, the total of 760 participants from four public senior high schools were enrolled in this study.

In this cross-sectional study, the correlation among school climate, perceived stigma, mental health literacy, and help-seeking behavior were explored. Prior to data collection of aforementioned variables, the students of four public senior high schools (i.e., SMAN 1 Gading, SMAN 1 Kraksaan, SMAN 1 Probolinggo, SMAN 1 Gending) were required to read and

complete the written informed consent. The written informed consent was received from all participants, which was duly authorized by their parents or legal representatives. A comprehensive document detailing the research study, encompassing the methodology, funding, data confidentiality and anonymity measures, result dissemination procedures, and information about the research team and their contact details, was provided and thoroughly explained to the participants. Three days after, we collected the signed informed consents and questionnaires about demographic variables, California school climate and safety survey-short form (school climates), mental health literacy scale (mental health literacy), peer mental health stigmatization scale (perceived stigma), and general help-seeking questionnaire (help-seeking behavior). The total duration of completing all questionnaires was 40 minutes with 1 time of 5 minutes' break. The participants were allowed to quit from the study in every times if they were not willing to complete all questionnaires. The researcher explained that this study was anonymous and would not affecting their school performance. In addition, a gift would be given to the participants who completed all questionnaires.

## Instruments

**California school climate and safety survey-short form (CSCSS-SF).** The California school climate and safety survey was developed by Furlong et al. in 2005 at Centre for School-Based Youth Development California [25]. The CSCSS-SF is a self-report assessment designed to evaluate general school climates and personal safety-related experiences [25]. The CSCSS-SF consisted three major domains, which were school danger, school climate, and victimization [25]. The school climate domain was generated to assess student's perceptions of school environment including support, respect, safety feeling, and interaction between individuals [25]. In this study, five Likert (from "1 = strongly disagree" to "5 = strongly agree") of school climate domain was used to capture school climate among adolescents in Indonesia. School climate domain contains total 13 items, which scored ranging between 13 to 65. The higher score indicating the student feels more positive school climate at school. This study also conducted reliability test and confirmatory analysis of school climate domain. The result showed school climate has acceptable reliability (i.e., 0.77) and good fit indices ($\chi^2$ = 125.742, df = 71, p < .001, Tucker–Lewis index [TLI] = 0.93, comparative fit index [CFI] = 0.95, and root mean square error of approximation [RMSEA] = 0.06), indicating school climate domain is a valid and reliable instrument to measure school climate among Indonesian adolescents.

**Mental health literacy scale.** The mental health literacy scale (MHLS) was established in Australia in 2015 by Matt O'Connor and Leanne Casey [26]. MHLS is a comprehensive mental health literacy measure comprising seven attributes: Capability to identify particular conditions; knowledge to gain the information; knowledge about the risk factors and triggers, knowledge of self-treatment; knowledge of available clinical help; and attitude that encourages recognition or effective help-seeking actions [26]. The MHLS contains 19 items with Likert scale responses. The item numbers 1–5 are rated on a four-point Likert scale from 1 (*very unlikely/unhelpful*) to 4 (*very likely/helpful*), and item numbers 6–19 are rated on a five-point Likert scale from 1 to 5 (1 = *strongly disagree/definitely unwilling* and 5 = *strongly agree/definitely willing*) [26]. The MHLS total score is generated by summing the scores of all items. Thus, the total score is ranging from 19 to 90 which higher scores indicated higher mental health literacy level. The MHLS was reliable and valid measurement to explore mental health literacy among adolescents, showed by 0.87 for Cronbach's alpha score and $\chi^2$ = 215.042, df = 137, $p$ < 0.0001, goodness-of-fit index = 0.91, TLI = 0.91, CFI = 0.92, RMSEA = 0.05, incremental fit index = 0.93 for construct validity indices which prior analyzed.

**Peer mental health stigmatization scale.**   Peer mental health stigmatization scale (PMHSS) was designed in Ireland to measure adolescent's related internal stigma and adolescents perception related their society stigma based on the stigma's components including stereotypes, prejudice, discrimination, and low social status [27]. The items in PMHSS were comprised two major part which negative and positive statements. The negative statement has two domains of stigma agreement (endorsing stigmatization statements on a personal level) and stigma awareness (recognition of the existing societal stigma) [27]. The interpretation of PMHSS was originally separated between negative and positive statement [27]. In this study, the 9 items related negative statement were used to identify the perceived stigma in Indonesian adolescents. The items are rated on a five-point Likert scale from 1 (*disagree completely*) to 5 (*agree completely*), and the scores are generated through addition all items, which range 9–45. Higher scores determine higher levels of perceived stigma. Based on our analysis, PMHSS was reliable and valid to be used in Indonesian adolescents. The reliability of PMHSS in Indonesian version was 0.78 and the construct validity indices were $\chi^2 = 38.90$ ($\chi^2/df = 1.62$), TLI = 0.93, CFI = 0.95, RMSEA = 0.05, SRMR = 0.03.

**General help-seeking questionnaire.**   The general help-seeking questionnaire (GHSQ) was created to investigate the help-seeking intentions of any sources and issues [28]. The GHSQ has been used in several studies and found there was a positive association between the general help-seeking and aspects of emotional competence [29, 30] in both prospective and retrospective behavior of help-seeking [31]. This help-seeking intention questionnaire includes formal and informal sources of help options aside from "self" or "no one" (e.g., counselor, peers, relatives, and no one). The GHSQ requests participants to rate their help-seeking intentions on a seven-point scale (1 = *extremely unlikely*, 7 = *extremely likely*) to each problem type for each source of help option including "no one." Higher scores indicate higher intentions to seek help [28]. The Cronbach's alpha of GHSQ was 0.86.

## Data analysis strategies

SPSS software for Windows (version 24.0; SPSS Inc., Chicago, IL, USA) was performed for statistical analyses. The descriptive statistics which are represented as the mean and standard deviation (SD) were calculated for demographic characteristic. Bivariate analysis was conducted for evaluating the relationship among demographic characteristics, school climate, perceived stigma, mental health literacy, and help-seeking behavior. T-test was used for categorical variables to evaluate the independent variables, and dependent variables were investigated using Pearson correlation for continuous variables.

The mediating roles of significant factors in the correlation between school climate and help-seeking behavior were evaluated. PROCESS SPSS (version 3.3; developed by Andrew F. Hayes) was used to analyze mediating roles [32]. This analysis was conducted using four models with a bootstrap of 5000 and a confidence interval (CI) of 95%. This study investigated the effects of the mediator on the correlation between independent and dependent variables after controlling for demographic characteristics. Fully mediated occurs only when the effect of the independent variable on the dependent variable is determined without controlling for the mediator [33]. Partially mediated occurs when X (independent variable) exerts a significant effect on Y (dependent variable) after controlling for the mediator [33].

## Ethical approval

This research was approved by the ethics research committee of the University in Malang on September 12, 2019 (approval number: 440/KEPK-POLKESMA/2019).

## Results

### Demographic data of participants

In Table 1, the demographic characteristics and mean score of school climate, perceived stigma, mental health literacy, and help-seeking behavior are listed. The participants' mean age was 16.55 years, with an SD of 0.68. More than half of the participants were female (71.4%; n = 543) and of Javanese ethnicity (84.1%; n = 639). The participants were distributed evenly across grades 10, 11, and 12, with percentages of 32.4%, 33.7%, and 33.9%, respectively, and almost all (99.3%; n = 755) had no repeated grade experience. In their school life, the majority of the participants (97.6%; n = 742) had close friends. Regarding family income, the participants mostly (40.1%; n = 305) belonged to a low-income family and had moderately educated parents. Senior high school was the highest education level of their fathers (44.6%; n = 339) as well as mothers (36.6%; n = 278). Furthermore, the health-related decision maker in their family was their mother among 58% (n = 441) of the participants. The mean score of the school climate of the participants was 46.42, with an SD of 6.14; for help-seeking behavior's mean score was 37.02 (SD = 11.31). Furthermore, the perceived stigma and mental health literacy of participants were investigated, and the resulting mean scores were 23.37 (SD = 4.78) and 58.17 (SD = 6.46), respectively.

### Mean score of help-seeking behavior

The mean scores of help-seeking behaviors for each variable and characteristic were given in Table 2. Variables namely ethnicity, family income, and father's education level were significant, with p-values less than .05 and .001. Among various ethnicities, Javanese, Madurese, and other ethnicities (Sundanese and Kalimantan) had mean scores of 37.46 (SD = 11.04), 34.16 (SD = 11.96), and 47.60 (SD = 17.39), respectively. Regarding family income, help-seeking behavior of participants from families with low income, moderate income, and high income had mean scores of 35.62 (SD = 11.42), 37.23 (SD = 11.15), and 38.61 (SD = 11.13), respectively. Finally, father's education level of all participants had a mean score ranging from 33.28 to 38.64 across all educational levels.

### Bivariate analysis among variables

To analyze the bivariate correlation among school climate, help-seeking behavior, mental health literacy, and perceived stigma, the Pearson correlation was performed. As shown in Table 3, school climate had a significant (p < .001, r = 0.146) correlation with help-seeking behavior. School climate and perceived stigma were significantly negatively correlated (p < .001, r = −0.130). However, the correlation between school climate and mental health literacy was nonsignificant. Among help-seeking behavior, mental health literacy, and perceived stigma, only the correlation between help-seeking behavior and mental health literacy was significant (p < .001, r = 0.102). Finally, in the last bivariate analysis, mental health literacy and perceived stigma were significantly negatively correlated with help-seeking behavior (p < .001, r = −0.208).

### Mediation model

To evaluate the multiple mediation model, the PROCESS macro for SPSS (Model 6) was performed. Fig 2 describe that not all of the pathways were statistically significant. Pathway 1 of "school climate → perceived stigma → help-seeking" was significant (indirect effect: −0.002; 95% CI: −0.042, −0.001). Pathway 2 of "school climate → mental health literacy → help-seeking behavior" was not significant (indirect effect: −0.001; 95% CI: −0.013, 0.012). Pathway 3 of

**Table 1. Demographic characteristics, school climate, perceived stigma, mental health literacy, and help-seeking behavior (n = 760).**

| Variables | N | % | M (SD)/Range |
|---|---|---|---|
| **Demographics** | | | |
| Age | 760 | 100.0 | 16.55 (0.68)/16–18 |
| Gender | | | |
| Male | 217 | 28.6 | |
| Female | 543 | 71.4 | |
| Grade levels | | | |
| Grade 10 | 246 | 32.4 | |
| Grade 11 | 256 | 33.7 | |
| Grade 12 | 258 | 33.9 | |
| Ethnicity | | | |
| Javanese | 639 | 84.1 | |
| Madurese | 116 | 15.3 | |
| Others | 5 | 0.7 | |
| Closest friends in school | | | |
| Yes | 742 | 97.6 | |
| No | 18 | 2.4 | |
| Repeating grade experience | | | |
| Yes | 5 | 0.7 | |
| No | 755 | 99.3 | |
| Family income (IDR/month) | | | |
| <1 million | 305 | 40.1 | |
| 1–2 million | 215 | 28.3 | |
| >2 million | 240 | 31.6 | |
| Decision maker in family | | | |
| Father | 319 | 42.0 | |
| Mother | 441 | 58.0 | |
| Father's educational levels | | | |
| Elementary school | 127 | 16.7 | |
| Junior high school | 90 | 11.8 | |
| Senior high school | 339 | 44.6 | |
| Bachelor | 153 | 20.1 | |
| Master | 37 | 4.9 | |
| Others (D1 and D3) | 14 | 1.8 | |
| Mother's educational levels | | | |
| Elementary school | 166 | 21.8 | |
| Junior high school | 144 | 18.9 | |
| Senior high school | 278 | 36.6 | |
| Bachelor | 142 | 18.7 | |
| Master | 17 | 2.2 | |
| Others (D1 and D3) | 13 | 1.7 | |
| School climate | | | 46.42 (6.14)/28–60 |
| Perceived stigma | | | 23.37 (4.78)/9–36 |
| Mental health literacy | | | 58.17 (6.46)/39–82 |
| Help-seeking behavior | | | 37.02 (11.31)/13–70 |

IDR, Indonesian rupiah; M, mean; SD, standard deviation

**Table 2. Help-seeking behavior score and the correlation (n = 760).**

| Variables | Mean | SD | p-value |
|---|---|---|---|
| Age | 37.02 | 11.307 | 0.58 |
| Gender | | | 0.09 |
| Male | 35.86 | 12.21 | |
| Female | 37.48 | 10.90 | |
| Grade levels | | | 0.34 |
| Grade 10 | 37.06 | 11.26 | |
| Grade 11 | 36.27 | 11.45 | |
| Grade 12 | 37.73 | 11.21 | |
| Ethnicity | | | 0.01* |
| Javanese | 37.46 | 11.04 | |
| Madurese | 34.16 | 11.96 | |
| Others | 47.60 | 17.39 | |
| Closest friends in school | | | 0.42 |
| Yes | 36.97 | 11.31 | |
| No | 39.17 | 11.32 | |
| Repeating grade experience | | | 0.23 |
| Yes | 40.40 | 09.15 | |
| No | 37.00 | 11.32 | |
| Family income (IDR/month) | | | 0.01* |
| <1 million | 35.62 | 11.42 | |
| 1–2 million | 37.23 | 11.15 | |
| >2 million | 38.61 | 11.13 | |
| Decision maker in family | | | 0.63 |
| Father | 37.26 | 11.46 | |
| Mother | 36.85 | 11.20 | |
| Father's educational levels | | | 0.01* |
| Elementary school | 33.28 | 10.80 | |
| Junior high school | 35.87 | 12.64 | |
| Senior high school | 37.81 | 11.27 | |
| Bachelor | 38.55 | 10.73 | |
| Master | 38.54 | 10.35 | |
| Others (D1 and D3) | 38.64 | 09.94 | |
| Mother's educational levels | | | 0.20 |
| Elementary school | 35.19 | 21.8 | |
| Junior high school | 36.60 | 18.9 | |
| Senior high school | 37.77 | 36.6 | |
| Bachelor | 37.72 | 18.7 | |
| Master | 39.65 | 2.2 | |
| Others (D1 and D3) | 37.92 | 1.7 | |

IDR, Indonesian rupiah; SD, standard deviation

*p < 0.05 Significant

"school climate → stigma → mental health literacy → help-seeking behavior" was significant (indirect effect: 0.004; 95% CI: 0.001, 0.010). Pathway 3 indicated that a high level of school climate was associated with low perceived stigma (b = −0.11, p < 0.001) and high mental health literacy (b = 0.28, p < 0.001) and help-seeking behavior (b = 0.14, p < 0.001). The perceived

**Table 3. Intercorrelations among the study variables.**

|  | School climate | Help-seeking behavior | Mental health literacy | Perceived stigma |
|---|---|---|---|---|
| **School climate** | 1 |  |  |  |
| **Help-seeking behavior** | 0.146** | 1 |  |  |
| **Mental health literacy** | 0.015 | 0.102** | 1 |  |
| **Perceived stigma** | −0.130** | 0.035 | −0.208** | 1 |

**r > 1

stigma and mental health literacy sequentially played partial mediating roles in the correlation between school climate and help-seeking behavior. The multiple mediation model accounted for at least 4.82% of the help-seeking behavior among adolescents in Indonesia ($R^2$ = 0.048).

## Discussion

To the best our knowledge, our study is the first to comprehensively explore the relationship among school climate, perceived stigma, mental health literacy, and help-seeking behavior in Indonesian adolescents. Furthermore, our study is the first to investigate whether perceived stigma and mental health literacy mediate the correlation between school climate and help-seeking behavior in Indonesian adolescents. Our study revealed a significant correlation between school climate and help-seeking behavior, school climate and perceived stigma, perceived stigma and mental health literacy, and mental health literacy and help-seeking behavior.

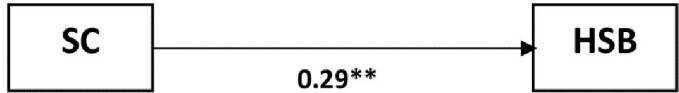

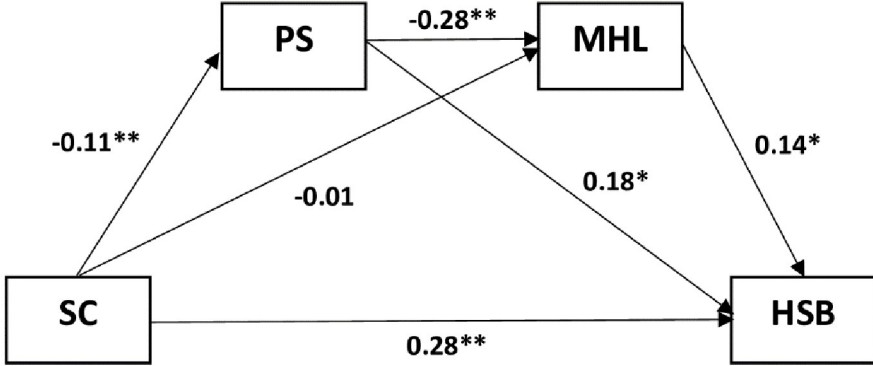

**Fig 2. Mediation model of the effect of perceived stigma and mental health literacy on the relationship between school climate and help-seeking behavior (n = 760).** Multiple mediation model described three indirect pathways. From the figure, we observed that Pathway I demonstrated a noteworthy indirect impact of PS on the relationship between SC and HSB. On the other hand, Pathway 2 revealed that MHL did not significantly mediate the association between SC and HSB. In the case of Pathway 3, it was found that PS and MHL sequentially played partial mediating roles in the correlation between SC and HSB. SC, school climate; HSB, help-seeking behavior; PS, perceived stigma; MHL, mental health literacy.

In addition, our study indicated the indirect relationship of school climate and help-seeking behavior through sequential mediating of perceived stigma and mental health literacy.

The present study result was consistent with a previous study that explained that supportive school climate improves help-seeking attitude in adolescents, for instance, students who perceive their teacher to be sympathetic, regardful, and concerned in them were likely to tell a teacher when they are facing a problem [16]. Similarly, theories of social interaction related to help seeking argue that the feeling of students related to potential helpers effect their decision making of help-seeking [34], and the stage-environment fit theory explains that human behavior is influenced by the characteristics of individuals and their environments [35] The correlation between the psychological needs of students and their school environment influences the motivation to seek help when facing a problem [36]. Another, according to a previous study the perception of positive and negative social consequences involving school climate is a critical factor influencing the willingness to use mental health services [37]. Two ways by which the school climate may affect help-seeking were described: (1) The willingness of students to admit to themselves that they have mental health problems and (2) their willingness to share those concerns to the supporting person and connect them with proper services in their school or community [38].

One particularly relevant of school based anti-stigma program may be school climate. It is considered in order to develop a sense of self-esteem, independence, and self-efficacy among adolescents [39]. Our finding showed that school climate was negatively associated with perceived stigma. According to Koth et al. [38], increase in mental health stigmatization among students in less supportive school climate. According to literature, social consequence school climate including peer, family, school staff, and other adults including has a role in stigma perception regarding mental help-seeking among adolescents [37]. Supportive school climate including excellent engagement (students, peers, teachers) and feeling safe encourage student to accept differences, prevent and reduce discrimination attitude [40]. Perceived stigma was highly co-incident with discrimination attitude [41]. Furthermore, the people with mental health problems was mostly marginalized in their society, however, not in supportive school climate [42]. We speculated, supportive school climate might reduce the discrimination attitude toward student with mental health problems, as consequence the public stigma in the school will be less. Thus, students in a positive school climate established low perceived stigma.

Our study showed no significant correlation between school climate and mental health literacy. In contrast to a previous study that was conducted among senior high school student, explained, a positive school climate was linked to mental health literacy [38]. According to a previous study, some of the predictors of mental health literacy were high education level, received services related to mental health, and lifetime experience with mental health disorders [43]. In the present study, we enrolled healthy participants, and therefore, no participant had mental health problems and this may influence the result of mental health literacy. It was opposite to the school climate; according to a previous study, a healthy person has a better perception regarding school climate than a person suffered from mental health problems, which may influence the significant association between school climate and mental health literacy [44].

The perceived stigma and mental health literacy was significantly associated in the present study. In the mediating model was showed that perceived stigma and mental health literacy was negatively associated, indicating lower perceived stigma will be associated with higher mental health literacy. In line with our finding, studies have described that perceived stigma was negatively correlated with mental health literacy [38, 45, 46]. Insufficient understanding regarding mental health problems can result in unfavorable attitudes and beliefs towards behavioral health, the services provided for it, and individuals with mental illness [47]. A

randomized control trial explained that improving mental health literacy through educational program effectively reduced the stigma of mental illness in adolescents [48]. These aforementioned literatures suggesting that perceived stigma could serve as crucial area in term of mental health literacy.

Some studies have shown that perceived stigma predicted help-seeking behavior [49, 50], which is different from our study's finding. In this study, perceived stigma was not significantly associated with help-seeking behavior. In line with this finding, a previous study by mentioned that perceived stigma was not found to be associated with less willingness to help-seeking, especially in delivery their mental health problems to the helper [51]. Furthermore, [51] noted that in the rural population, the help seeking was not associated with perceived stigma due to in rural populations tend to cope with their mental health problems by themselves [51]. The present study enrolled part of its participants from rural areas, which may have influenced the correlation between perceived stigma and help-seeking behavior in this study due to tend to cope up with their mental health problems by themselves. In addition, family income significantly associated with help-seeking behavior and the highest percentage of participants in our study was from low-income families, that might be the obstacle to seek help for their mental health problem. Prior longitudinal study described no significant association between perceived stigma and help-seeking behavior, its due to some variables that may confound the relationship was not observed [20]. It was consistent with our finding, perceived stigma significantly associated with help-seeking behavior after simultaneously analyzed in our mediation model. We assumed some factors such as family income affected the association of perceived stigma and help-seeking behavior. Another point of view, a study mentioned that personal stigma was more prominent barrier compared to perceived stigma for young people seeking help [52]. Young people who confident about their treatment or behavior might be less concern of the attitude of most people (public stigma) such friends and family.

Mental health literacy was significantly correlated to help-seeking behavior in our study. A pilot study was conducted in first year psychology students (17–26 years old) mentioned that mental health literacy score was achieved from students determining the intention of seeking help when suffering from mental health problem [53]. Furthermore, other studies suggest that mental health literacy become one of vital predictors for help-seeking and a modifiable factor to improve help-seeking behavior, moreover intention in joining an educational interventions [54, 55]. The literation about mental health problems might influence the perception and decision of adolescence to emerge the intention for seeking help. Besides, a study explained that knowledge and beliefs regarding mental health problems will assist to recognition, treatment and prevention [56]. Thus, our study's finding was consistent that mental health literacy positively associated to help-seeking behavior.

Our serial mediation model revealed that perceived stigma and mental health literacy sequentially mediated the relationship between school climate and help-seeking behavior. The coefficient in the model showed positive association was observed between school climate and mental health literacy and help-seeking behavior, however, negatively associated to perceived stigma. This finding indicated that supportive school climate will reduce perceived stigma, then, increase the mental health literacy to improve help-seeking behavior among adolescent. In contrast with prior study, social and self-stigma was described to have significantly partial mediating effect on relationship mental health literacy and help-seeking attitudes [57]. Higher mental health literacy indicate an adequate knowledge and beliefs of young people to aid to destigmatize and manage mental health problems event [57]. This different finding was assumed due to vice versa relationship of perceived stigma and mental health literacy. These two variables was important to determine help-seeking behavior. The recognition of mental health problems could be achieved by adequate mental health literacy and the stigma could

affect the decision making to take the action for seeking-help [58]. Reversely, large number of stigma from their society intend to inhibit adequate mental health literacy then followed by unidentified mental health problem and ended to inadequate help-seeking. In our study, indirect relationship was also observed in the mediation effect of perceived stigma on the relationship between school climate and help-seeking behavior, but not for mental health literacy. However, the indirect effect of perceived stigma mediation model only reached 2%. Supported by previous studies that revealed, students who encircled by a positive school climate showed fewer stigmatized beliefs and higher rates of mental health literacy than those who live among less positive schools climates [38], which ultimately increased the help-seeking behavior among adolescents [37, 38]. Our serial mediation model explained a comprehensives mechanism of help-seeking behavior model by a supportive school climate lead to decreasing perceiving stigma and inclined mental health literacy, then improve the help-seeking behavior among adolescents.

## Limitations

This study has several limitations that need to be considered. First, we used a cross-sectional design and thus reporting correlations and not causal effects. Second, our measurement tools were self-reported which provide subjective information, and may result in interpretation bias. Finally, we collected data from four senior high schools only representing a subset of senior high schools across the country. In view of these limitations, we conclude that our findings cannot be generalized nationally.

## Conclusions

Our study indicated partial roles in sequential mediating model of perceived stigma and mental health literacy in the relationship between school climate and help-seeking behavior in Indonesian adolescents. Thus, our study discovered novel insight for mental health provision that to improve help-seeking behavior among adolescent, a supportive school climate is necessary. Furthermore, health provision also need to consider student's perceived stigma and mental health literacy as important factors to achieve adequate help-seeking behavior.

## Supporting information

**S1 Raw Data.**
(XLSX)

## Acknowledgments

The authors would like to thank Wallace Academic Editing for their editing services and our colleague Rathi Paramastri for English editing.

## Author Contributions

**Formal analysis:** Mohammad Hendra Setia Lesmana, Min-Huey Chung.

**Methodology:** Mohammad Hendra Setia Lesmana.

**Writing – original draft:** Mohammad Hendra Setia Lesmana, Min-Huey Chung.

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
