## [Decision Letter · Decision Letter 0]

28 Mar 2023

PONE-D-22-22187Mediating roles of perceived stigma and mental health literacy in the relationship between school climate and help-seeking behavior in Indonesian adolescentsPLOS ONE

Dear Dr. Chung,

Thank you for submitting your manuscript to PLOS ONE. After careful consideration, we feel that it has merit but does not fully meet PLOS ONE’s publication criteria as it currently stands. Therefore, we invite you to submit a revised version of the manuscript that addresses the points raised during the review process.

We look forward to receiving your revised manuscript.

Kind regards,

Azizollah Arbabisarjou, Ph.D

Academic Editor

PLOS ONE

Journal Requirements:

Additional Editor Comments (if provided):

Dear Author/s

The authors have to correct the paper as Honor Reviewer Yasutaka Ojio (Reviewer 2) have suggested " Major Revision".

In next , I evaluate the paper and make a decision.

Reviewers' comments:

Reviewer's Responses to Questions

**Comments to the Author**

1. Is the manuscript technically sound, and do the data support the conclusions?

Reviewer #1: No

Reviewer #2: Partly

2. Has the statistical analysis been performed appropriately and rigorously? 

Reviewer #1: No

Reviewer #2: I Don't Know

3. Have the authors made all data underlying the findings in their manuscript fully available?

Reviewer #1: No

Reviewer #2: No

4. Is the manuscript presented in an intelligible fashion and written in standard English?

Reviewer #1: No

Reviewer #2: Yes

5. Review Comments to the Author

Reviewer #1: Line 53: What is a “sufficiently early age”?

Line 58-59: One sentence has adolescence listed as 16-24. The previous sentence says 15-19. Adolescence does not extend into the 20-24 period. This needs to be addressed.

Line 62: What previous survey? This is unclear, add specific mention of the author or source if you want to use this phrasing

Some of the introduction is repetitive and needs to be revised (e.g., stats regarding 10% of adolescents experiencing a MH problem is described twice). A lot of the statistics presented are redundant and could be condensed into one paragraph.

Most of the intro is just statistics on mental health and treatment in adolescence. There is barely any space dedicated to establishing and discussing the variables that will ultimately be used in the analysis. A better review of the literature is needed with respect to mental health literacy and perceived stigma. How is school climate defined? How does that fit into what the authors are testing? Terms need to be operationally defined and hypotheses need to be presented. Essentially the authors are arguing for a mediational model, but that model is not set up in the introduction at all. A figure is needed with the hypothesized paths highlighted, and the intro needs to establish the theoretical basis for testing that model.

The power analysis that was conducted is only for a regression analysis, but the authors tested a mediational model. The power analysis of a mediational model should be examining specific paths and the indirect effects in the model.

I still don’t know what school climate is supposed to be measuring. There is mention of fit statistics for a model of school climate, but no description of the analysis and model building process that was used to obtain those fit statistics. I assume there was a CFA performed, but that is not stated explicitly anywhere in the paper. The fit statistics are only adequate here.

Same goes for the peer stigmatization scale. Where are these fit statistics from? Are these based on current data?

The authors appear to have used PROCESS but that is not stated (there is a citation, but it should be stated explicitly)

I don’t think table 4 is useful. All of the paths and indirect effects can be presented in the figure.

As this is a cross-sectional study, it is very difficult to argue for mediation when it is possible that the order of variables could be swapped. Have alternative models been explored? What happens when you switch the order of perceived stigma and mental health literacy? Cross-sectional mediation analyses can be justified when there is a strong case for the temporal precedence of the variables, or a strong theoretical argument for setting up the model the way it has been set up. Neither of those is true in this case, and not enough work was done in the introduction to support the model that was tested.

Discussion now introduces demographics as a variable of interest? Again the variables in the analysis need to be established in the introduction with supporting literature.

There is a substantial body of literature establishing the relationship between perceived stigma and help-seeking. But in the present study, that relationship is not significant. The authors argue it may be because of rural participants in the study, but this is a fairly weak argument, unless a substantial proportion of the participants were rural.

Some of the literature cited in the discussion should be in the intro to help establish why the variables of interest were studied, and to develop an argument for the model tested.

The paragraph on mental health literacy and perceived stigma doesn’t make any connections to the existing study. How do the findings of the present study relate to those presented in this paragraph?

A lot of the discussion is just summaries of previous research without connection to the study results.

The discussion needs to give space to alternative models.

The discussion mentions a parallel model, but only one model was tested from what is presented. What is the parallel model?

The manuscript has a lot of grammatical errors throughout that sometimes impact clarity and understanding. The manuscript needs to be carefully edited.

Reviewer #2: Thank you for the opportunity to review this manuscript. The authors examined the relationship between the key keywords ``School climate'', ``Mental health literacy'', ``Stigma'', and ``Help-seeking'' in recent mental health challenges in adolescents. Investigating and clarifying these relationships is an essential process in creating an approach to creating a better environment for adolescents in the real world. I would like to ask for some consideration for international researchers and practitioners who are active in this field.

1. Detailed definitions of the keywords ``School climate'', ``Mental health literacy'', ``Stigma'', and ``Help-seeking'' are important to the reader's understanding. For example, the meaning of mental health literacy differs greatly depending on mental well-being, mental ill-health, or focus on treatment and care.

2. Hypotheses about the causal direction of the associations of the variables shown by the authors are not convincing enough in the current manuscript. In particular, a scientific-based theory, etc. should be presented before analyzing why perceived stigma and mental health literacy as mediating roles factors in the relationship between SC and HSB.

3. The results in the abstract require show statistical information.

4. Line 56 (Introduction section)

Are the authors referring to "suicide"? Please add information with several references.

5. I found some similar terms like mental health issues, mental issues, and mental problems. Please provide a clear definition of each.

6. Does Indonesia have a School-based mental health curriculum? Does each school have a staff or care unit taking mental health?

7. Line 114 (Methods section)

The authors described that participants were recruited by convenience sampling. For example, a detailed explanation of the informed consent procedure is required.

8. Line 321 (Descussion section), the direction of Stigma and mental health literacy (Knowledge) is controversial. I would like a careful discussion.

6. PLOS authors have the option to publish the peer review history of their article (what does this mean?). If published, this will include your full peer review and any attached files.

Reviewer #1: No

Reviewer #2: No

---

## [Author Response · Author response to Decision Letter 0]

20 Jul 2023

Reviewer #1: 

1. Q: Line 53: What is a “sufficiently early age”?

A: Thank you for the question, we recognized that this sentence has ambiguous meaning. We revised this sentence into:

 “Around 70% of adolescents suffering from mental problem did not receive appropriate treatment immediately” [Line 65-67]. 

2. Q: Line 58-59: One sentence has adolescence listed as 16-24. The previous sentence says 15-19. Adolescence does not extend into the 20-24 period. This needs to be addressed.

A: We really appreciate your concern and we agree that the sentences above were conflicting. Thus, we modified the sentences as: 

“The occurrence of mental health problems was highest among individuals in the 16-24 age group compared to any other period in life [2]. Mental health problems were also frequently observed during childhood and adolescence, impacting about 14% of individuals aged 4 to 17 years [3].” [Line 53-56].

3. Q: Line 62: What previous survey? This is unclear, add specific mention of the author or source if you want to use this phrasing

A: Thank you for your comment, the previous survey in the sentence refers to ASEAN mental health system report. We revised the sentence as: 

“According to the ASEAN mental health system report in 2016, a significant number of mental health problems were prevalent among adolescents in Indonesia” [Line 62-63].

4. Q: Some of the introduction is repetitive and needs to be revised (e.g., stats regarding 10% of adolescents experiencing a MH problem is described twice). A lot of the statistics presented are redundant and could be condensed into one paragraph.

A: Thank you for your comment, the introduction was revised accordingly as:

“Mental health problems have become a global burden, particularly among adolescents [1]. The occurrence of mental health problems was highest among individuals in the 16-24 age group compared to any other period in life [2]. Mental health problems were also frequently observed during childhood and adolescence, impacting about 14% of individuals aged 4 to 17 years [3]. Approximately, 1 in 10 adolescents was affected by mental health problems, including anxiety, depression, and personality disorder [4]. In Indonesia, the Ministry of Education, Culture, Research, and Technology (KEMENDIKBUD) implemented Policy Number 111 in 2014 in response to the growing concerns about mental health issues among adolescents [5]. This policy mandates that every school should establish a dedicated department or section focused on guidance and counseling to promote the overall well-being of their students [5]. However, according to the ASEAN mental health system report in 2016, a significant number of mental health problems were prevalent among adolescents in Indonesia [6]. The 2013 Indonesia Basic Health Research (RISKESDAS) reported that mental health problems were accounted by 11.3% in East Java, Indonesia [7]. In addition, around 70% of adolescents suffering from mental health problems did not receive appropriate treatment immediately [4]. The World Health Organization (WHO) reported that 10% to 20% of adolescents encountered by mental health problems remained underdiagnosed and undertreated [8]. Meanwhile, a study mentioned that 35% of adolescents who experiencing emotional and mental health problems did not tend to seek help [9]. Furthermore, only one in three adolescents with mental health problems sought for help [10]. Mental health problems among adolescents contributed to suicide and self-harm that ranking as the fourth leading cause of mortality in the world [8], as well as, the top 10 cause of disabilities in East Java, Indonesia [11]. Therefore, we conclude that mental health problems in adolescents is a serious concern, particularly, regarding help-seeking behavior among adolescents.

According to reasoned action theory, people's behavior is expected to be rational, consistent, and frequently automatic, based on their beliefs regarding the behavior they are performing [12]. There are three factors affected people’s behavior [12]. First, background factors related information such as knowledge. Second, people’s beliefs including behavioral belief such as stigma that followed by attitude toward the behavior. Third, the behavior can be directly influenced by factors of actual control, such as the environment. The existence of environmental limitations can hinder individuals from engaging in specific behavior including help-seeking behavior [12]. Based on studies, adolescents in Asia spend most of their time at school, which makes school has pivotal roles in maintaining adolescents mental well-being [13-15]. In this regard, the school climate was pronounced highly associated with help-seeking behavior. The supportive school environment from teachers, peers, and safety environment increased the willingness of adolescents to express their concern and accessing help [16-18]. In addition, positive school climate (i.e., safety, engagement and environment aspects) engaged adolescents to increase literacy and decrease stigma about mental health problem [19]. Moreover, prior studies suggested that perceived or internal stigma on mental health problem would inhibit adolescents to seek for help [9, 20, 21]. On the other hand, higher mental health literacy level encouraged adolescents seeking for help [22]. Based on aforementioned explanation, the association among environment (school climate), knowledge (mental health literacy), behavioral belief (perceived stigma), and behavior (help-seeking behavior) need to be explored further (see Figure 1). 

To the best our knowledge, there was no study conducted to explore the link among school climate, mental health literacy, perceived stigma, and help-seeking behavior in adolescents. Furthermore, lacking study related help-seeking behavior was conducted in Indonesian adolescents. Thus, our study aimed to analyze the mediating effect of mental health literacy and perceived stigma on the relationship between school climates and help-seeking behavior in Indonesian adolescents.” 

[Line 52-101].

5. Q: Most of the intro is just statistics on mental health and treatment in adolescence. There is barely any space dedicated to establishing and discussing the variables that will ultimately be used in the analysis. A better review of the literature is needed with respect to mental health literacy and perceived stigma. How is school climate defined? How does that fit into what the authors are testing? Terms need to be operationally defined and hypotheses need to be presented. Essentially the authors are arguing for a mediational model, but that model is not set up in the introduction at all. A figure is needed with the hypothesized paths highlighted, and the intro needs to establish the theoretical basis for testing that model.

A: Thank you for your concern. We added some information in the introduction as below. 

“According to reasoned action theory, people's behavior is expected to be rational, consistent, and frequently automatic, based on their beliefs regarding the behavior they are performing [11]. There are three factors affected people’s behavior [11]. First, background factors related information such as knowledge. Second, people’s beliefs including behavioral belief such as stigma that followed by attitude toward the behavior. Third, the behavior can be directly influenced by factors of actual control, such as the environment. The existence of environmental limitations can hinder individuals from engaging in specific behavior including help-seeking behavior [11]. Based on studies, adolescents in Asia spend most of their time at school, which makes school has pivotal roles in maintaining adolescents mental well-being [12-14]. In this regard, the school climate was pronounced highly associated with help-seeking behavior. The supportive school environment from teachers, peers, and safety environment increased the willingness of adolescents to express their concern and accessing help [15-17]. In addition, positive school climate (i.e., safety, engagement and environment aspects) engaged adolescents to increase literacy and decrease stigma about mental health problem [18]. Moreover, prior studies suggested that perceived or internal stigma on mental health problem would inhibit adolescents to seek for help [8, 19, 20]. On the other hand, higher mental health literacy level encouraged adolescents seeking for help [21]. Based on aforementioned explanation, the association among environment (school climate), knowledge (mental health literacy), behavioral belief (perceived stigma), and behavior (help-seeking behavior) need to be explored further (see Figure 1).” [Line 76-95]

Fig 1. Hypothesis Framework Based on Reasoned Action Model and Related Literatures 

6. Q: The power analysis that was conducted is only for a regression analysis, but the authors tested a mediational model. The power analysis of a mediational model should be examining specific paths and the indirect effects in the model.

A: Thank you for your comment. For sample size calculation for mediating analysis, we follow sample paper to achieve statistical significant result (https://doi.org/10.1111/ijn.12164) . We explained detail in method part as below: 

“The sample size was calculated under statistical test of multiple linear regression analysis with 12 predictors, including demographic variables, school climate, perceived stigma, and mental health literacy. The effect size of 0.15 [23], Cronbach’s alpha of 0.05, and a power of 0.95 were inputted. the sample size was calculated equals to 184 participants. After considering the dropout rate of 20%, this study at least recruited 184+37= 221 participants. However, the total of 760 participants from four public senior high schools were enrolled in this study.” [Line 114-120]

A: For the indirect effects in the model, we explained in result part as below: 

“To evaluate the multiple mediation model, the PROCESS macro for SPSS (Model 6) was performed. Figure 1 describe that not all of the pathways were statistically significant. Pathway 1 of “school climate � perceived stigma � help-seeking” was significant (indirect effect: −0.002; 95% CI: −0.042, −0.001). Pathway 2 of “school climate � mental health literacy � help-seeking behavior” was not significant (indirect effect: −0.001; 95% CI: −0.013, 0.012). Pathway 3 of “school climate � stigma � mental health literacy � help-seeking behavior” was significant (indirect effect: 0.004; 95% CI: 0.001, 0.010). Pathway 3 indicated that a high level of school climate was associated with low perceived stigma (b = −0.11, p < .001) and high mental health literacy (b = −0.28, p < .001) and help-seeking behavior (b = 0.14, p < .001). The perceived stigma and mental health literacy sequentially played partial mediating roles in the correlation between school climate and help-seeking behavior. The multiple mediation model accounted for at least 4.82% of the help-seeking behavior among adolescents in Indonesia (R2 = 0.048).” [Line 270-282]

7. Q: I still don’t know what school climate is supposed to be measuring. There is mention of fit statistics for a model of school climate, but no description of the analysis and model building process that was used to obtain those fit statistics. I assume there was a CFA performed, but that is not stated explicitly anywhere in the paper. The fit statistics are only adequate here.

A: Thank you, we really appreciate your comment. We realized that school climate measurement was not explained clearly. We revised it in the method section as below:

“The California school climate and safety survey (CSCSS-SF) was developed by Furlong et al. in 2005 at Centre for School-Based Youth Development California [25]. The CSCSS-SF is a self-report assessment designed to evaluate general school climates and personal safety-related experiences [25]. The CSCSS-SF consisted three major domains, which were school danger, school climate, and victimization [25]. The school climate domain was generated to assess student’s perceptions of school environment including support, respect, safety feeling, and interaction between individuals [25]. In this study, five Likert (from “1=strongly disagree” to “5=strongly agree”) of school climate domain was used to capture school climate among adolescents in Indonesia. School climate domain contains total 13 items, which scored ranging between 13 to 65. The higher score indicating the student feels more positive school climate at school. This study also conducted reliability test and confirmatory analysis of school climate domain. The result showed school climate has acceptable reliability (i.e., 0.77) and good fit indices (ᵡ2 = 125.742, df = 71, p < .001, Tucker–Lewis index [TLI] = 0.93, comparative fit index [CFI] = 0.95, and root mean square error of approximation [RMSEA] = 0.06), indicating school climate domain is a valid and reliable instrument to measure school climate among Indonesian adolescents.” [Line 142-157]

8. Q: Same goes for the peer stigmatization scale. Where are these fit statistics from? Are these based on current data?

A: Thank you for your question, we conducted construct validity prior to mediating analysis using a part of current data. We revised the paragraph in method section as below: 

“Peer mental health stigmatization scale (PMHSS) was designed in Ireland to measure adolescent’s related internal stigma and adolescents perception related their society stigma based on the stigma’s components including stereotypes, prejudice, discrimination, and low social status [27]. The items in PMHSS were comprised two major part which negative and positive statements. The negative statement has two domains of stigma agreement (endorsing stigmatization statements on a personal level) and stigma awareness (recognition of the existing societal stigma) [27]. The interpretation of PMHSS was originally separated between negative and positive statement [27]. In this study, the 9 items related negative statement were used to identify the perceived stigma in Indonesian adolescents. The items are rated on a five-point Likert scale from 1 (disagree completely) to 5 (agree completely), and the scores are generated through addition all items, which range 9-45. Higher scores determine higher levels of perceived stigma. Based on our analysis, PMHSS was reliable and valid to be used in Indonesian adolescents. The reliability of PMHSS in Indonesian version was 0.78 and the construct validity indices were χ2 = 38.90 (χ2/df = 1.62), TLI = 0.93, CFI = 0.95, RMSEA = 0.05, SRMR = 0.03. ” [Line 175-189]

9. Q: The authors appear to have used PROCESS but that is not stated (there is a citation, but it should be stated explicitly)

A: We are really appreciate your comment, we revised it accordingly as below.

“The mediating roles of significant factors in the correlation between school climate and help-seeking behavior were evaluated. PROCESS SPSS (version 3.3; developed by Andrew F. Hayes) was used to analyze mediating roles [32].” [Line 209-211]

10. Q: I don’t think table 4 is useful. All of the paths and indirect effects can be presented in the figure.

A: Thank you for your suggestion, we deleted the table 4 as your concern.

11. Q: As this is a cross-sectional study, it is very difficult to argue for mediation when it is possible that the order of variables could be swapped. Have alternative models been explored? What happens when you switch the order of perceived stigma and mental health literacy? Cross-sectional mediation analyses can be justified when there is a strong case for the temporal precedence of the variables, or a strong theoretical argument for setting up the model the way it has been set up. Neither of those is true in this case, and not enough work was done in the introduction to support the model that was tested.

A: Thank you very much for your comment. We revised and added sentences in introduction part

---

## [Decision Letter · Decision Letter 1]

17 Jan 2024

Mediating roles of perceived stigma and mental health literacy in the relationship between school climate and help-seeking behavior in Indonesian adolescents

PONE-D-22-22187R1

Dear Dr. Chung,

We’re pleased to inform you that your manuscript has been judged scientifically suitable for publication and will be formally accepted for publication once it meets all outstanding technical requirements.

Kind regards,

Muhammad Arsyad Subu, Ph.D

Academic Editor

PLOS ONE

Additional Editor Comments (optional):

Reviewers' comments:

Reviewer's Responses to Questions

**Comments to the Author**

1. If the authors have adequately addressed your comments raised in a previous round of review and you feel that this manuscript is now acceptable for publication, you may indicate that here to bypass the “Comments to the Author” section, enter your conflict of interest statement in the “Confidential to Editor” section, and submit your "Accept" recommendation.

Reviewer #2: (No Response)

Reviewer #3: All comments have been addressed

2. Is the manuscript technically sound, and do the data support the conclusions?

Reviewer #2: Yes

Reviewer #3: Yes

3. Has the statistical analysis been performed appropriately and rigorously? 

Reviewer #2: Yes

Reviewer #3: I Don't Know

4. Have the authors made all data underlying the findings in their manuscript fully available?

Reviewer #2: No

Reviewer #3: Yes

5. Is the manuscript presented in an intelligible fashion and written in standard English?

Reviewer #2: Yes

Reviewer #3: Yes

6. Review Comments to the Author

Reviewer #2: To Authors:

Thank you for responding to my comments. I think the paper has significantly improved. The additional information you provided raised further consideration.

1. The authors provided a more detailed explanation of the scales used in the study regarding "school climate", "mental health literacy", "stigma", and "help-seeking" as essential variables in this study. The authors also explained the research hypothesis using figures. In the manuscript, they explained it as followings, in Lines 92-95. “Based on aforementioned explanation, the association among environment (school climate), knowledge (mental health literacy), behavioral belief (perceived stigma), and behavior (help-seeking behavior) need to be explored further (Fig 1).” Is your current explanation fully consistent with the measures used in this study? For example, The mental health literacy scale (MHLS) by O'Connor et al. 2015 includes elements other than knowledge. Although it is not necessary to have an additional analysis or reanalyze the part of the factors of MHLS, the authors need to be more detailed in describing their hypotheses in the introduction section. In addition, this amendment may make the authors improve the discussion section further.

2. The authors wrote the following in their previous response letter, but it seems that the corrections have not been reflected in the entire text. Please check and revise it. “We used the term “mental health problems”. Additionally, to make our manuscript more consistent, we revise the mentioned term (i.e., mental health issues, mental issues, and mental problems) into mental health problems.”

Reviewer #3: Remove some terms which is not internationally recognised or translate fully in English such as KEMDIKBUD and RISKESDAS. Translate into English some words such as SMAN 1 Gading and so on.

the study conducted prior to covid-19 pandemic which may be different from the current situation. I think the authors need to analyze the different situation and how the study meaningful in the current situation.

7. PLOS authors have the option to publish the peer review history of their article (what does this mean?). If published, this will include your full peer review and any attached files.

Reviewer #2: No

Reviewer #3: No

---

## [Editor Report · Acceptance letter]

7 May 2024

PONE-D-22-22187R1 

PLOS ONE

Dear Dr. Chung, 

I'm pleased to inform you that your manuscript has been deemed suitable for publication in PLOS ONE. Congratulations! Your manuscript is now being handed over to our production team.

Kind regards, 

on behalf of

Dr. Muhammad Arsyad Subu 

Academic Editor

PLOS ONE